# Snow-Ice-Inspired Approach for Growth of Amorphous Silicon Nanotips

**DOI:** 10.3390/nano9050680

**Published:** 2019-05-02

**Authors:** Seungil Jo, Hyunsoo Kim, Nae-Man Park

**Affiliations:** 1School of Semiconductor and Chemical Engineering, Semiconductor Physics Research Center, Chonbuk National University, 567, Baekje-daero, Deokjin-gu, Jeonju 54896, Korea; jsipower@jbnu.ac.kr; 2Materials and Components Laboratory, Electronics and Telecommunications Research Institute (ETRI), 218, Gajeong-no, Yuseong-gu, Daejeon 34129, Korea; 3ETRI School, University of Science and Technology, 217, Gajeong-no, Yuseong-gu, Daejeon 34113, Korea

**Keywords:** amorphous silicon nanotips, aqueous solution, low-temperature solution process, liquid–solid mechanism

## Abstract

The growth of one-dimensional nanostructures without a metal catalyst via a simple solution method is of considerable interest due to its practical applications. In this study, the growth of amorphous silicon (*a*-Si) nanotips was investigated using an aqueous solution dropped onto the Si substrate, followed by drying at room temperature or below for 24 h, resulting in the formation of *a*-Si nanotips on the Si substrate. Typically, the *a*-Si nanotips were up to 1.6 μm long, with average top and middle diameters of 30 and 80 nm, respectively, and contained no metal catalyst in their structure. The growth of *a*-Si nanotips can be explained in terms of the liquid–solid mechanism, where the supercritical Si solution (liquid) generated on the Si substrate (after reaction with the aqueous solution) promotes the nucleation of solid Si (acting as seeds) on the roughened surface, followed by surface diffusion of Si atoms along the side wall of the Si seeds. This is very similar to the phenomenon observed in the growth of snow ice crystals in nature. When photoexcited at 265 nm, the *a*-Si nanotips showed blue luminescence at around 435 nm (2.85 eV), indicating feasible applicability of the nanotips in optoelectronic functional devices.

## 1. Introduction

The growth of one-dimensional (1D) nanostructures, such as nanowires (NWs), nanorods, and nanotubes, is of considerable interest in terms of its potential applications in novel functional electrical and optical devices, including light-emitting diodes, field-effect transistors, solar cells, and sensors [1]. Therefore, multiple approaches have been introduced to grow 1D nanomaterials using various materials such as Si [2,3], C [4,5], ZnO [6,7], GaN [8,9], TiO_2_ [10,11], SiOx [12,13], and so forth. Among these 1D nanomaterials, Si NWs have been the most widely studied because Si is the most common industrial semiconductor and nanomaterials can facilitate the design of novel functional devices beyond the confines of Si films [14,15]. One-dimensional Si nanomaterials are conventionally grown via metal catalytic growth using a chemical solution. However, this method has critical drawbacks because it requires high temperatures (>1000 °C) in a vacuum chamber [15] or pressures above 200 bar in an autoclave [16]. Top-down fabrication of Si NWs using plasma or chemical etching techniques have recently been reported [17,18]. However, these methods also entail complex processing and equipment or require temperatures above 300 °C. To overcome these problems, we recently demonstrated a novel aqueous-solution-based method for growing crystalline Si (c-Si) NWs at relatively low temperatures (85 °C) under atmospheric pressure [19]. The basic concept of our method was the use of a water-soluble Si precursor (sodium methyl-siliconate, SMS) and the addition of potassium iodide (KI) for activating the Au catalyst (because KI etches Au and thereby enhances catalyst activity at low process temperatures). Consequently, the Si monomers obtained from decomposed precursors could effectively react with the Au nanoseeds, resulting in effective formation of c-Si NWs. Indeed, this process was very simple and cost-competitive compared with the conventional solution method.

In this study, we further developed a simple growth technique to obtain amorphous Si (*a*-Si) nanotips via one of the formation mechanisms of snow ice crystals. Specifically, dropping an aqueous solution, comprising SMS, KI, and gold nanoparticles (Au NPs), onto the Si substrate, accompanied by solution droplet cooling, resulted in the formation of *a*-Si nanotips. The growth of *a*-Si nanotips could be explained by the liquid–solid mechanism. The *a*-Si nanotips showed photoluminescence (PL) at around 435 nm, indicating that their applications in optoelectronic functional devices are feasible.

## 2. Materials and Methods 

To make an aqueous solution, KI powder (0.002 mol, Sigma-Aldrich, Seoul, Korea) and 10 mL of Au NPs (3 nM, particle size = 20 nm, Sigma-Aldrich, Seoul, Korea) dispersed in H_2_O were mixed in 80 mL of deionized water. Subsequently, 20 mL of 30% aqueous SMS (Gelest, Morrisville, PA, USA) was added to the mixed solution. In order to mix the solution properly, the aqueous solution was stirred magnetically at 85 °C on a hot plate. Then, samples were prepared by dropping aqueous solution onto the Si substrate. All samples were cooled down 4 °C or room temperature (RT), or maintained at 70 °C over 24 h until the solution droplets dried. The structural, compositional, and optical properties of the samples were investigated using scanning electron microscopy (SEM, SU-8230, Hitachi, Japan), energy-dispersive X-ray spectrometry (EDX, SU-8230, Hitachi, Japan), X-ray diffraction (XRD, X’pert Pro Powder, PANalytical, Netherlands), transmission electron microscopy (TEM, HD-2300A, Hitachi, Japan), and PL (SpectraPro 500i, Acton, USA). To further investigate the growth mechanism of the *a*-Si nanotips on the Si substrate, focused ion beam (FIB)-SEM (FB-2100, Hitachi, Japan) and atomic force microscopy (AFM, MOD-1M series, Nanofocus, Richmond, VA, USA) analyses were also performed.

## 3. Results

For the growth of the *a*-Si nanotips, we used an aqueous-solution-based process (Figure 1a). The main idea of this study was to just drop the aqueous solution onto the Si substrate. The SEM images of the samples (Figure 1b) showed that several nanotips were formed on the Si substrate. The magnified SEM image (Figure 1c) showed that the nanotips were typically up to 1.6 μm long with average top and middle diameters of ~30 and ~80 nm, respectively. They showed a unique morphology, namely, sharp tips with a smaller diameter through the end, and they also contained no metallic catalyst. The EDX spectrum of the nanotips revealed that the structures were composed mainly of Si and a little O, with no trace of impurity elements, indicating that the nanotips were made of Si and O, as shown in Figure 1d. In addition, the XRD pattern (Figure 1e) showed only one diffraction peak for the (100) Si substrate, while other diffraction peaks were not observed. This means that the Si nanotips had an amorphous structure.

To further investigate the structural properties of the *a*-Si nanotip, TEM measurement was performed, as shown in Figure 2. To prepare the sample for TEM analysis, the *a*-Si nanotips were selectively scraped onto the TEM grid and were immobilized by dispersion in acetone. The TEM images showed that no metal catalyst was present at the ends of the *a*-Si nanotip, as shown in Figure 2a. The *a*-Si nanotip was also found to have top and middle diameters of 32 and 76 nm, respectively, and a length of ~1.5 μm, which were in excellent agreement with the SEM observation. The corresponding EDX elementary mapping images also revealed that the *a*-Si nanotip was mainly composed of Si and O atoms (Figure 2b). The selected area electron diffraction (SAED) pattern of *a*-Si nanotips showed a typical diffusive ring pattern (Figure 2c), which means that the Si nanotip was amorphous. According to the TEM analysis, it was confirmed that the Si nanotips had an amorphous structure and the main constituent elements were Si and O.

In order to elucidate the growth mechanism of the *a*-Si nanotips, FIB-SEM analysis was performed. The cross-sectional SEM image clearly showed well-grown *a*-Si nanotips and a ~100-nm-thick *a*-Si wetting layer between the *a*-Si nanotips and Si substrate, as shown in Figure 3a. According to our EDX analysis (Figure 3b), the wetting layer was primarily composed of Si. To investigate the interfaces of the *a*-Si wetting layer and the Si substrate, AFM surface analysis was performed, as shown in Figure 3c. For this measurement, the sample grown with *a*-Si nanotips and a wetting layer was dipped into the buffered oxide etchant to remove the top *a*-Si layers. For comparison, the bare surface of Si substrate chemically etched was also measured. The surface roughness obtained from an area of 3 × 3 μm^2^ was as high as 30 nm for the *a*-Si nanotip sample, while it was 0.2 nm for the bare Si substrate. Therefore, the interface of the *a*-Si wetting layer and the Si substrate showed a rougher or more undulating surface. This indicates that the chemical reaction between the Si substrate and aqueous solution caused substantial surface roughening associated with surface etching. This is reasonable because the hydroxide ions (OH^−^) originating from the aqueous solution under a strongly basic condition can etch the surface of a Si substrate [19].

Based on these findings, the so-called liquid–solid mechanism was proposed as the growth process of *a*-Si nanotips, as shown in Figure 4. For example, the aqueous solution dropped on the Si substrate provided OH^−^ groups that could dissolve the Si substrate. Thereby, the Si substrate surface was roughened and the supersaturated droplets with dissolved Si entirely covered the Si substrate. With aging, solidification can occur at the roughened Si surface (acting as nucleation sites), resulting in the generation of *a*-Si seeds. Especially, KI used as an Au etchant sufficiently increases the surface energy and reactivity of Au as a metal catalyst for the formation of *a*-Si seeds even at very low temperatures [19]. In a supercritical fluid environment, the nanotip can be grown via surface diffusion of Si atoms along the side wall of the *a*-Si seeds. This is known as “Berg’s effect”, in which surface diffusion of Si atoms is driven by the concentration gradient of the Si atoms along the side wall of the nanotips from the bottom to the top [20]. This mechanism indicates why the *a*-Si nanotip had a sharp tip-like structure. A similar growth phenomenon was previously reported for the tapered ZnO nanostructure (made from liquid Zn) by Yuan et al. [21], where the growth temperature was high (over 400 °C). This behavior at low temperatures is observed in natural snow ices, in which high condensation towards the top surface leads to the formation of needle-like ice crystals [22,23].

According to our proposed growth mechanism for *a*-Si nanotips, the dissolution of Si atoms from the substrate is indispensably in need of a supercritical environment. To check indirectly if our proposed mechanism is correct, our process was performed on a sapphire substrate. In this case, no Si nanostructures were observed. This indicates that the chemical etching reaction between the aqueous solution and Si substrate was the most important step for the supersaturation of liquid Si resulting in the growth of *a*-Si nanotips. In addition, for the growth of *a*-Si nanotips using the liquid–solid mechanism, the nucleation and upward surface diffusion should occur sequentially. The low temperature is also greatly valuable to actualize a supercritical environment. Figure 5 indicates that the structure of *a*-Si nanotips can be controlled by changing the cooling temperature rather than KI concentration (i.e., the lower the cooling temperature, the longer and sharper are the *a*-Si). This result is plausible considering that both nucleation and surface diffusion processes strongly depend on a cooling temperature. The density of *a*-Si nanotips was about 1 ± 0.5 E9/cm^2^ in the condition of KI 0.002 mL.

Figure 6 shows the RT PL spectrum from *a*-Si nanotips obtained under excitation at 265 nm. For this measurement, the *a*-Si nanotips were solely scraped onto the substrate, on which the PL measurement was performed. A stable and strong PL was observed at around 435 nm (2.85 eV) with a shoulder peak at 414 nm (3 eV). Nishikawa et al. investigated the presence of PL bands ranging from 1.9 to 4.3 eV for high-purity silica glasses and observed PL peaks at around 2.7 and 3.0 eV, which were attributable to the neutral O vacancy and intrinsic diamagnetic defects like two-fold-coordinated Si lone pair centers [24,25]. Therefore, the PL of our sample was also considered to originate from O deficiencies and/or intrinsic defects.

## 4. Conclusions

To summarize, we developed a novel method to grow *a*-Si nanotips on a Si substrate at RT or below under atmospheric pressure. The simple dropping of an aqueous solution onto the Si surface accompanied by cooling led to the formation of high-density *a*-Si nanotips. The growth mechanism could be understood by the liquid–solid mechanism, where the nucleation of Si seeds at the roughened surface followed by upward surface diffusion along the side walls of the seeds formed the *a*-Si nanotips. Blue luminescence was also observed for the *a*-Si nanotips under excitation at 265 nm, indicating that the nanotips have potentially promising applications in optoelectronic devices.

## Figures and Tables

**Figure 1 nanomaterials-09-00680-f001:**
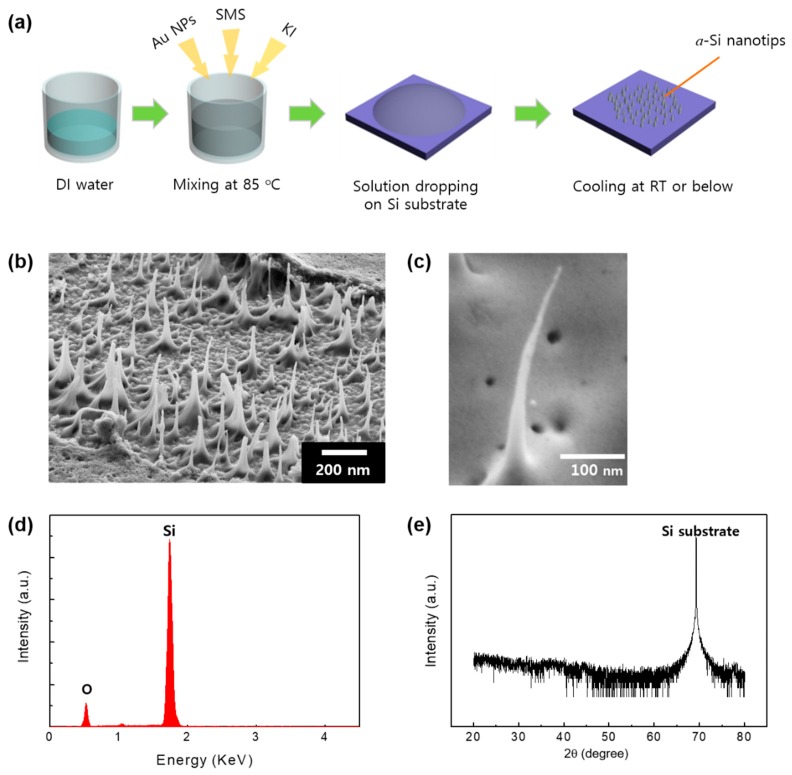
(**a**) Schematic growth procedure of amorphous Si (*a*-Si) nanotips on a Si substrate. (**b**) SEM image of the sample surface. (**c**) Magnified scanning electron microscopy (SEM) image and (**d**) corresponding energy-dispersive X-ray spectrometry (EDX) spectrum of a single *a*-Si nanotip. (**e**) X-ray diffraction (XRD) pattern of *a*-Si nanotips.

**Figure 2 nanomaterials-09-00680-f002:**
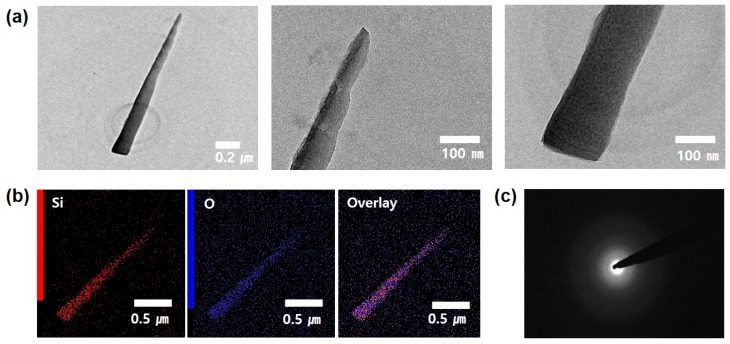
(**a**) Transmission electron microscopy (TEM) images, (**b**) EDX elementary mapping, and (**c**) selected area electron diffraction (SAED) pattern of a single *a*-Si nanotip.

**Figure 3 nanomaterials-09-00680-f003:**
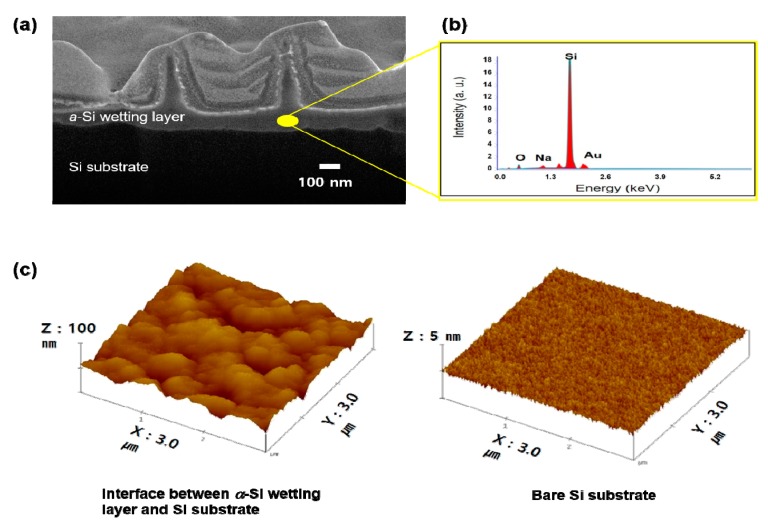
(**a**) Focused ion beam (FIB)-SEM image of the *a*-Si nanotips on a silicon substrate. (**b**) EDX spectrum of *a*-Si wetting layer at the yellow point in (**a**). (**c**) 3 × 3 μm^2^ atomic force microscopy (AFM) images of the interface between the *a*-Si wetting layer and Si substrate (left) and the surface of a bare Si substrate (right).

**Figure 4 nanomaterials-09-00680-f004:**
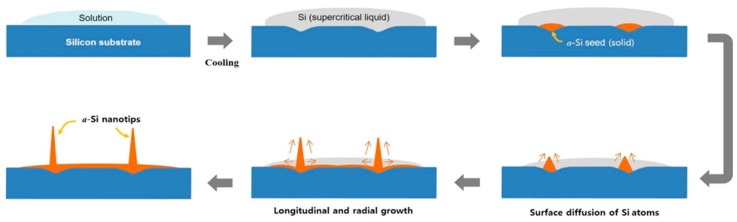
Schematic illustration of *a*-Si nanotip growth.

**Figure 5 nanomaterials-09-00680-f005:**
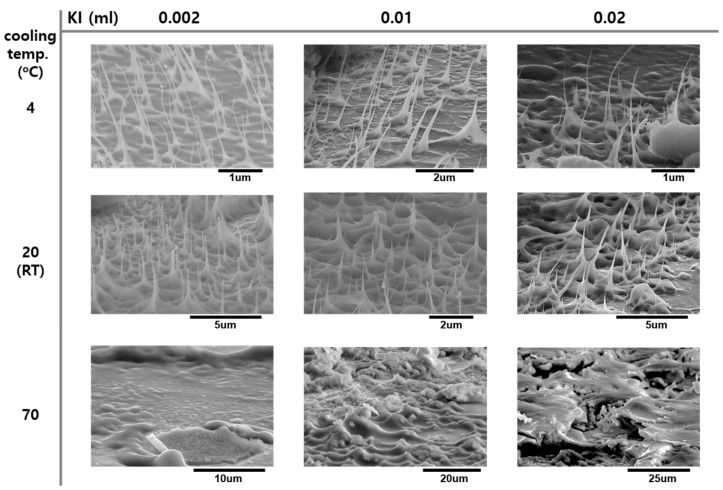
SEM images of *a*-Si nanotips formed on a Si substrate as functions of KI concentration and cooling temperatures.

**Figure 6 nanomaterials-09-00680-f006:**
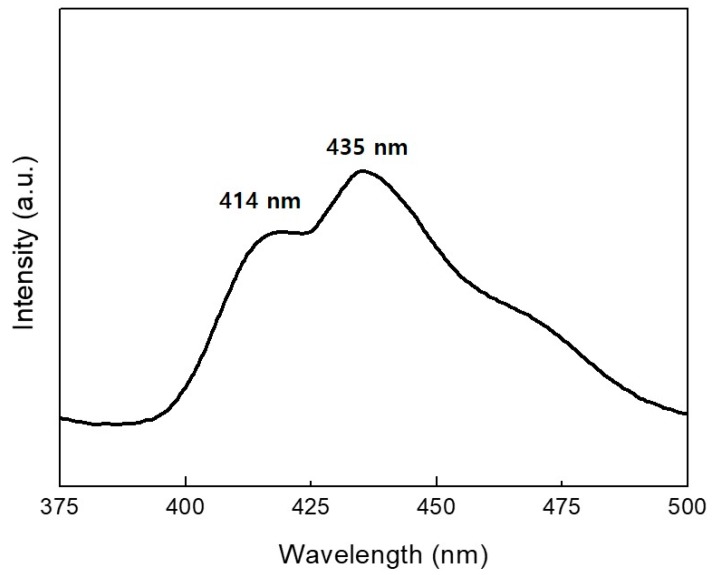
Room temperature (RT) photoluminescence (PL) spectrum of the *a*-Si nanotips obtained under excitation at 266 nm.

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
