# Peer review of "Snow-Ice-Inspired Approach for Growth of Amorphous Silicon Nanotips"

_nanomaterials, 2019, doi:10.3390/nano9050680_

Reviewer 1 Report

The paper is simple, interesting and well written.

I have some minor revisions:

1. It is necessary to insert the marker in the SEM images of figure 5.

2. I would like to have a density of Silicon tips, fundamental point for applications.

3. The PL originates from defects; it would be nice to test reproducibility.

The introduction is too poor, there are several works in the literature focused on chemical etching of silicon without the use of temperature  and the gold contamination to produce silicon NWs with a density of 10 12 / cm 2.

For example, the authors could introduce these references:

Nature nanotechnology 9 (1), 19,2014

Nanoscale 4 (9), 2863-2866, 2012

ACS sensors 3 (9), 1690-1697,2018

Author Response

A1. Figure 5 was corrected with scale bars

A2. The density was about 1(±0.5) E9 cm-2 which was added on 161st lines on page 6.

A3. When We made many samples and observed PL, some did not show PL, but 70~80% of samples showed the reproducible PL.

A4. According to the comment, the referred paper was inserted in the reference list.

Reviewer 2 Report

Interesting work and well presented. However, this work along with your prior work (ref 19) states this as "novel" aq solution-based SiNW synthesis. However, There is other low T & P solution synthesis of SiNW such as that of Korgel et al. J Am Chem Soc 2008, 130, 5436-5437. Comparing to this work (or similar work) would be good for reader to recognize novelty better.

Your EDX of Si nanotip shows no Au (Fig 1d) but Fig 3d shows Au at the base. Adding a comment in text about this would be good.

More minor text edits. In line 131 "however, was high over 400 C" is awkward and should be simplified. And line 168 states "promising application" is a bit strong statement based on one PL spectrum. I see it as potentially promising.

Author Response

A1. The EDX spectrum in Fig. 1 is about Si nanotip, but that in Fig. 3 is about Si wetting layer. Au exists in the Si wetting layer, but there is no Au in Si nanotips.

A2. According to the comments, the sentences was corrected.

Reviewer 3 Report

In this manuscript by Jo, Kim and Park the authors have described a method of growing nanotips from silicon. The reported findings are well supported with experimental data and the provided analysis does not require further rectification. I have only three small remarks, which would possibly improve this work:

1) AuNPs - please give more details regarding the synthesis of AuNPs used for the study. At present, only the concentration and diameter is reported.

2) Figure 3 - it is distorted. Please correct it.

3) Figure 5 - please include professional scale bars (not those generated by the microscope with redundant information at the bottom).

Author Response

A1. More information about Au NPs was added in the Materials and Methods section.

A2. I did not understand your comment. If you describe it in more detail, I will correct it. Furthermore, SEM and AFM images in Fig. 3 have scale bars and, therefore, it seems to have no problem.

A3. According to the comment, scale bars were added.

Round  2

Reviewer 3 Report

Thank you for your response. All the concerns but one (the most important) have been taken care of. You really need to provide some meaningful information how you obtained these NPs e.g. synthesis parameters or name of the supplier if you got them from commercial sources. This is absolutely crucial to make this study reproducible by others. Without this information the study is nice and interesting, but not very useful - I am afraid.

Author Response

Dear Reviewer

I am sorry for giving deficient information about Au NPs which was purchased from Sigma-Aldrich.

The reason that I did not pay attention to this information is based on my experience.

Word is, I grew c-Si NWs with synthesized Au NPs and  commercially available ones and the results from these two conditions were almost same. In these cases, NP concentration, volume, and particle size were same. So, I ignored the information, but I added more information in the Materials and Methods section according to your comment.

Round  3

Reviewer 3 Report

Thank you very much. I can now approve this interesting work for publication.